# Asynchronous Thruster Fault Detection for Unmanned Marine Vehicles under DoS Attacks

Fuxing Wang
*School of Automation Engineering*
*University of Electronic Science and Technology of China*
Chengdu 611731, China
wfx614328@163.com

Yue Long
*School of Automation Engineering*
*University of Electronic Science and Technology of China*
Chengdu 611731, China
longyue@uestc.edu.cn

Tieshan Li
*School of Automation Engineering*
*University of Electronic Science and Technology of China*
Chengdu 611731, China
tieshanli@126.com

*Abstract*—This paper investigates a thruster fault detection strategy for unmanned marine vehicles (UMVs) subjected to external disturbances and aperiodic Denial of Service (DoS) attacks. To address the challenge of timely detection of DoS attacks, the UMV and the corresponding filters are modeled within the framework of an asynchronous switched system. Sufficient conditions ensuring the system's exponential stability and prescribed performance are derived using model-dependent average dwell time and piecewise Lyapunov functions. Additionally, the tolerable lower bound of the sleep interval and the upper bound of the attack interval for DoS attacks are established. Solvable conditions for the designed fault detection filters are obtained by leveraging decoupling techniques. Finally, simulations conducted on a UMV validate the effectiveness of the proposed methods.

*Index Terms*—Unmanned marine vehicles, asynchronous switched system, DoS attacks, fault detection.

## I. INTRODUCTION

In recent years, unmanned marine vehicles (UMVs) have attracted significant attention in marine science and technology due to their wide-ranging applications in marine exploration, environmental monitoring, and resource development [1]. Nevertheless, the operational environment for UMVs is inherently complex , and their reliance on wireless communication networks for communication with shore-based centers makes them vulnerable to external disturbances, equipment malfunctions, cyber-attacks, and other disruptions [2]. The unpredictable nature of potential harm caused by these disturbances or faults, combined with the inherent vulnerabilities of cyberspace, renders UMV systems particularly susceptible to cyber-attacks. These risks can result in system failures and potentially catastrophic accidents [3]. As a result, improving the reliability and security of UMVs has emerged as a crucial area of research and development.

This work is supported in part by the National Natural Science Foundation of China under Grants 62273072, 51939001. (*Corresponding author: Yue Long*)

The unpredictable nature of potential harm caused by disturbances or faults to unmanned marine vehicles (UMVs) underscores the critical need for a real-time fault detection (FD) warning mechanism. The core of fault detection methodology involves comparing system performances to identify fault signals. Current research predominantly focuses on model-based fault detection, which has shown significant success in various systems, including continuous-discrete systems [4], T-S fuzzy systems [5], and Markovian jump systems [6]. The primary approach involves generating residual signals through filters or observers and subsequently establishing a fault warning mechanism. For UMVs, several studies have made noteworthy contributions. [7] has explored the design of controllers and FD filters based on observers for networked UMVs, [8] proposed event-triggered fault detection mechanisms for UMVs in networked environments, and [2] utilized T-S fuzzy systems to model UMV systems, particularly addressing fault detection under replay attacks. Despite these advancements, the scope of fault detection research for UMVs remains relatively narrow and lacks comprehensive coverage [9]. Consequently, further investigation into robust and holistic fault detection strategies for UMVs is imperative to enhance their reliability and operational safety [10].

On the other hand, due to the openness of cyberspace, UMV systems are particularly vulnerable to cyber-attacks. Deception attacks and Denial of Service (DoS) attacks are currently common types of attacks [11]. Deception attacks involve sending incorrect or tampered data to the system [12], including replay attacks [13] and false data injection attacks [14]. Compared to deception attacks, DoS attacks cause signal transmission to be unavailable for a period, leaving the system in an open-loop state, which makes it easier to cause severe disruption in system operations. Consequently, numerous studies on DoS attacks have emerged [15], [16].

However, most existing research assumes that Denial of Service (DoS) attacks can be detected promptly, suggesting that the switching of filters corresponding to each subsystem

happens simultaneously with the subsystem switching [10], [17]. However, in practical applications, detecting DoS attacks in a timely manner proves challenging, leading to delays. This delay implies that the filter often takes additional time to adjust to the appropriate control mode based on the subsystem mode, resulting in asynchronous filter/subsystem switching [18]. As a result, filters designed for synchronous switching may not provide optimal detection performance in real-world scenarios [19]. Thus, incorporating asynchronous switching into thruster fault detection for unmanned marine vehicles (UMVs) under DoS attacks is of substantial practical significance.

Inspired by the previous discussion, this paper investigates thruster fault detection (FD) for unmanned marine vehicles (UMVs) under Denial of Service (DoS) attacks using an asynchronous switched method to enhance reliability and security. Addressing the challenge of timely DoS attack detection, the paper proposes an asynchronous switched filter specifically designed for thruster fault detection. Furthermore, leveraging model-dependent average dwell time (MDADT) and piecewise Lyapunov functions (PLF), the paper establishes the tolerable lower bound of the sleep interval and the upper bound of the attack interval for DoS attacks. The filter parameters are determined based on linear solvability conditions. The effectiveness of the proposed method is ultimately validated through simulation.

## II. PROBLEM FORMULATION AND MODELING

### A. UMV Model

Consider the UMV and the following body-fixed equations of motion

$$M\dot{\delta}(t) + N\delta(t) + R\psi(t) = E\varphi(t),$$
$$\dot{\psi}(t) = J(\eta(t))\delta(t), \tag{1}$$

where $\delta(t) = [\delta_u(t), \delta_v(t), \delta_r(t)]^T$ with $\delta_u(t), \delta_v(t), \delta_r(t)$ representing the surge, sway and yaw velocities, respectively. $\psi(t) = [x_p(t), y_p(t), \eta(t)]^T$ with $x_p(t)$ and $y_p(t)$ are positions and $\eta(t)$ is the yaw angle. $\varphi(t)$ is the control input. $M$, $N$, $R$ and $E$ denote inertia, damping, mooring forces and configuration matrices, and $M$ is a symmetric positive-definite and invertible matrix that satisfies $M = M^T > 0$, $J(\eta(t)) = \begin{bmatrix} cos(\eta(t)) & -sin(\eta(t)) & 0 \\ sin(\eta(t)) & cos(\eta(t)) & 0 \\ 0 & 0 & 1 \end{bmatrix}$.

Then, by defining $x(t) = \delta(t) - \delta_{ref}$, $A(t) = -M(t)^{-1}N(t)$, $B_1(t) = M(t)^{-1}R$ and $B_2(t) = M(t)^{-1}E$, and taking into account the unavoidable disturbance $\tilde{d}(t)$ caused by wind, wave and current, the system (1) can be expressed as

$$\begin{cases} \dot{x}(t) = Ax(t) + B_1d(t) + B_2\varphi(t), \\ y(t) = Cx(t), \end{cases} \tag{2}$$

where $d(t) = B_1(t)^{-1}d^*(t) - \psi(t) + B_1(t)^{-1}A\delta_{ref}$ and $C = \begin{bmatrix} 0 & 0 & 1 \end{bmatrix}$ denotes the output matrix.

Consider thruster fault $\varphi^F(t) = \rho\varphi(t) + \sigma f(t)$ and assume control inputs $\varphi(t) = Kx(t)$ are designed, (2) is represented as

$$\begin{cases} \dot{x}(t) = \hat{A}x(t) + B_1d(t) + B_2\hat{f}(t), \\ y(t) = Cx(t), \end{cases} \tag{3}$$

where $\hat{A} = A + B_2K$ and $\hat{f}(t) = -\bar{\rho}\varphi(t) + \sigma f(t)$.

### B. DoS Attacks Model

Consider the aperiodic dos attacks as follows:

$$A_{Dos} = \begin{cases} 0, & t \in [t_{2l}, t_{2l+1}) \triangleq \kappa_{0,2l} \\ 1, & t \in [t_{2l+1}, t_{2(l+1)}) \triangleq \kappa_{1,2l} \end{cases} \tag{4}$$

where $t \in [t_{2l}, t_{2l+1}) \triangleq \kappa_{0,2l}$ $(l \in N, t_{2l} \geq 0)$ indicates the $l^{th}$ sleep interval with the length $s_l = t_{2l+1} - t_{2l}$, and $t \in [t_{2l+1}, t_{2(l+1)}) \triangleq \kappa_{1,2l}$ indicates the $l^{th}$ DoS attacks interval with the length $d_l = t_{2(l+1)} - t_{2l+1}$.

Due to the communication disruption caused by DoS attacks, the UMV system (3) can be augmented into the following switched system, which has been discretized. The sleeping interval can be expressed as $k \in [k_{2l}, k_{2l+1})$, and the DoS attacks interval can be expressed as $k \in [k_{2l+1}, k_{2(l+1)})$.

$$\begin{cases} x(k+1) = A_{id}x(k) + B_{1id}d(k) + B_{2id}\hat{f}(k) \\ y(k) = C_dx(k) \end{cases}. \tag{5}$$

### C. Asynchronous Switching Filter

In the case of the DoS attacks and thruster faults, the residual signal produced by the switched filter is as follows:

$$\begin{cases} x_f(k+1) = A_{fi}x_f(k) + B_{fi}y(k) \\ r(k) = C_{fi}x_f(k) + D_{fi}y(k) \end{cases} \quad (i = 0, 1) \tag{6}$$

where $x_f(k)$ is the state of the filters, $r(k)$ is the residual signal of the switched system (5). Define $\tilde{x}(k) = \begin{bmatrix} x^T(k) & x_f^T(k) \end{bmatrix}^T$, $\varpi(k) = \begin{bmatrix} d^T(k) & f^T(k) \end{bmatrix}^T$ and the residual evaluation signal $e(k) = r(k) - \hat{f}(k)$, (6) is rewritten as (7)

$$\Phi_0 : \begin{cases} \tilde{x}(k+1) = \tilde{A}_i\tilde{x}(k) + \tilde{B}_i\varpi(k) \\ e(k) = \tilde{C}_i\tilde{x}(k) + \tilde{D}_i\varpi(k) \end{cases}, k \in [k_l + \varepsilon_l, k_{l+1})$$

$$\Phi_1 : \begin{cases} \tilde{x}(k+1) = \tilde{A}_{ij}\tilde{x}(k) + \tilde{B}_{ij}\varpi(k) \\ e(k) = \tilde{C}_{ij}\tilde{x}(k) + \tilde{D}_{ij}\varpi(k) \end{cases}, k \in [k_l, k_l + \varepsilon_l)$$
$$\tag{7}$$

where $i \neq j$, $i \in \{0, 1\}$, $j \in \{0, 1\}$, $\tilde{A}_{ij} = \begin{bmatrix} A_{id} & 0 \\ B_{fj}C_d & A_{fj} \end{bmatrix}$, $\tilde{B}_{ij} = \begin{bmatrix} B_{1i} & B_{2i} \\ 0 & 0 \end{bmatrix}$, $\tilde{C}_{ij} = \begin{bmatrix} D_{fj}C_d & C_{fj} \end{bmatrix}$ and $\tilde{D}_{ij} = \begin{bmatrix} 0 & -I \end{bmatrix}$.

To better set the stage for the next section, the following definitions are presented.

**Definition 1:** For any switching signal $\tau(k)$ and $0 < k_0 \leq k$, let $\mathcal{M}_{\tau,l}(k_0, k)$ indicate the number of switching times that the $l_{th}$ subsystem is activated over $[k_0, k)$. If

$$\mathcal{M}_{\tau,l}(k_0, k) \leq N_{\mathcal{M}_{0,l}} + \frac{N_l(k_0, k)}{\lambda_l}$$

holds for scalar $\lambda_l > 0$ and integer $N_{M_{0,l}} \geq 0$, then $\lambda_l$ is called model-dependent average dwell time. $N_l(k_0, k)$ is the total running time of the $l_{th}$ subsystem over $[k_0, k)$.

**Definition 2:** Consider asynchronous switched subsystems $\Phi_0$ and $\Phi_1$, and given scalar $\alpha$, $\beta$, and $\gamma$ satisfying $0 < \alpha < 1$, $\beta > 0$ and $\gamma > 0$. Under zero initial condition, if the asynchronous switched system is exponentially stable and satisfies $\sum_{s=k_0}^{\infty} (1-\alpha)^s e^T(s) e(s) \leq \gamma^2 \sum_{s=k_0}^{\infty} \varpi^T(s) \varpi(s)$, it is said that the system exhibits exponential stability and has exponential $H_\infty$ index $\gamma$.

## III. MAIN RESULTS

In this section, the stability and $H_\infty$ performance of asynchronous switched systems (7) will be analyzed, and the sufficient and linearly solvable conditions for the designed switched FD filters are given.

**Theorem 1:** Consider the switched subsystems $\Phi_0$ and $\Phi_1$ under DoS attacks, scalars $\alpha_i$, $\beta_i$, $\gamma$, $\mu_0$ and $\mu_1$ satisfying $0 < \alpha_i < 1$, $\beta_i > 0$, $\gamma > 0$, $\mu_0 > 1$ and $0 < \mu_1 < 1$, if there exist symmetric positive-definite matrices $\mathcal{P}_i$ satisfying the following conditions

$$\tilde{A}_i^T \mathcal{P}_i \tilde{A}_i - \mathcal{P}_i + \alpha_i \mathcal{P}_i < 0, \tag{8}$$

$$\tilde{A}_{ij}^T \mathcal{P}_i \tilde{A}_{ij} - \mathcal{P}_i - \beta_i \mathcal{P}_i < 0, \tag{9}$$

$$\mathcal{P}_i \leq \mu_i \mathcal{P}_j, \tag{10}$$

$$\tau_D < \frac{\varepsilon_M \ln \phi_1 + \ln \mu_1}{\ln \tilde{\alpha}_1}, \tau_F > -\frac{\varepsilon_M \ln \phi_0 + \ln \mu_0}{\ln \tilde{\alpha}_0}, \tag{11}$$

the switched subsystems $\Phi_0$ and $\Phi_1$ are exponentially asymptotically stable with the exponential $H_\infty$ performance, where $i \neq j$, $\tilde{\alpha}_i = 1 - \alpha_i$, $\tilde{\beta}_i = 1 + \beta_i$, $\phi_i = \frac{\tilde{\beta}_i}{\tilde{\alpha}_i}$ and $\varepsilon_M$ denotes the maximum time that the filter lags the subsystem.

*Proof:* The piecewise Lyapunov function for the closed-loop switched subsystems $\Phi_0$ and $\Phi_1$ are given as follows

$$\mathcal{V}_i(\tilde{x}(k)) = \tilde{x}^T(k) \mathcal{P}_i \tilde{x}(k). \tag{12}$$

When $\varpi(k) = 0$ and $k \in [k_{2l}, k_{2l+1})$, it can be obtained

$$\mathcal{V}(\tilde{x}(k)) \leq \begin{cases} \tilde{\alpha}_i^{k-k_{2l}-\varepsilon_{2l}} \mathcal{V}_i(\tilde{x}(k_{2l} + \varepsilon_{2l})), k \in \Gamma^+ \\ \tilde{\beta}_i^{k-k_{2l}} \mathcal{V}_i(\tilde{x}(k_{2l})), k \in \Gamma^- \end{cases}, \tag{13}$$

where $\tilde{\alpha}_i = 1 - \alpha_i$ and $\tilde{\beta}_i = 1 + \beta_i$. And when $k \in \mathcal{T}^+(k_{2l}, k_{2l+1})$, from (8) and (11), it can be derived

$$\begin{aligned} &\mathcal{V}(\tilde{x}(k)) \leq \tilde{\alpha}_0^{k-k_{2l}-\varepsilon_{2l}} \mathcal{V}_0(\tilde{x}(k_{2l} + \varepsilon_{2l})) \\ &\leq \tilde{\alpha}_0^{k-k_{2l}-\varepsilon_{2l}} \cdot \tilde{\beta}_0^{\varepsilon_{2l}} \cdot \mathcal{V}_0(\tilde{x}(k_{2l})) \\ &\leq \cdots \\ &\leq \theta \exp \left\{ \max \left( \frac{\varepsilon_M \ln \phi_0 + \ln \mu_0}{\tau_F} + v_0, -\frac{\varepsilon_M \ln \phi_1 + \ln \mu_1}{\tau_D} + v_1 \right) \right. \\ &\left. (\Xi_F(k_0, k) + \Xi_D(k_0, k)) \right\} \mathcal{V}(\tilde{x}(k_0)) \end{aligned} \tag{14}$$

where $\theta = \exp\left[ (\varepsilon_M \ln \phi_0 + \ln \mu_0) \xi_F - (\varepsilon_M \ln \phi_1 + \ln \mu_1) \xi_D \right]$, $\omega = \max \left\{ -\frac{\varepsilon_M \ln \phi_0 + \ln \mu_0}{\tau_F} - \ln \tilde{\alpha}_0, \frac{\varepsilon_M \ln \phi_1 + \ln \mu_1}{\tau_D} - \ln \tilde{\alpha}_1 \right\}$, $\chi_0 = \theta_0^{\varepsilon_M} \mu_0$, $\chi_1 = \theta_1^{\varepsilon_M} \mu_1$, $v_i = \ln \tilde{\alpha}_i$.

From (11), it has $\omega > 0$. Then, it is clear that $\mathcal{V}(\tilde{x}(k))$ converges to zero when $k \to \infty$. Therefore, the closed-loop switched subsystems $\Phi_0$ and $\Phi_1$ are exponentially asymptotically stable when (8) and (11) hold.

Next, if $\varpi(k) \neq 0$ for $k \in [k_{2l}, k_{2l+1})$ and zero initial conditions, (??) is derived as follows

$$\Delta \mathcal{V}_i(\tilde{x}(k)) < \begin{cases} -\alpha_i \mathcal{V}_i(\tilde{x}(k)) - \Upsilon(k), k \in \Gamma^+ \\ \beta_i \mathcal{V}_i(\tilde{x}(k)) - \Upsilon(k), k \in \Gamma^- \end{cases} \tag{15}$$

where $i = 0, 1$, $\Upsilon(k) = e^T(k) e(k) - \gamma^2 \varpi^T(k) \varpi(k)$. When $k \in \mathcal{T}^+(k_{2l}, k_{2l+1})$, it can have the following inequality in the similar way from (10) and (15)

$$\begin{aligned} \mathcal{V}(\tilde{x}(k)) \leq & \tilde{\alpha}_0^{k-k_{2l}} \tilde{\alpha}_0^{k_{2l-1}-k_{2l-2}} \cdots \tilde{\alpha}_0^{k_1-k_0} \phi_0^{\varepsilon_{2l}} \phi_0^{\varepsilon_{2l-2}} \cdots \phi_0^{\varepsilon_0} \cdot \\ & \mu_0^{M_F(k_0,k)} \tilde{\alpha}_1^{k_{2l}-k_{2l-1}} \cdots \tilde{\alpha}_1^{k_2-k_1} \phi_1^{\varepsilon_{2l-1}} \cdots \phi_1^{\varepsilon_1} \cdot \\ & \mu_1^{M_D(k_0,k)} \mathcal{V}(\tilde{x}(k_0)) - \tilde{\alpha}_0^{k-k_{2l}} \tilde{\alpha}_0^{k_{2l-1}-k_{2l-2}} \cdots \\ & \tilde{\alpha}_0^{k_1-k_0} \phi_0^{\varepsilon_{2l}} \phi_0^{\varepsilon_{2l-2}} \cdots \phi_0^{\varepsilon_0} \mu_0^{M_F(k_0,k)} \tilde{\alpha}_1^{k_2-k_{2l-1}} \cdots \\ & \tilde{\alpha}_1^{k_2-k_1} \phi_1^{\varepsilon_{2l-1}} \cdots \phi_1^{\varepsilon_1} \mu_1^{M_D(k_0,k)} \sum_{s=k_0+\Delta_0}^{k_1-1} \tilde{\alpha}_0^{k_1-s-1} \Upsilon(s) \\ & - \tilde{\alpha}_0^{k-k_{2l}} \tilde{\alpha}_0^{k_{2l-1}-k_{2l-2}} \cdots \tilde{\alpha}_0^{k_1-k_0} \phi_0^{\varepsilon_{2l}} \phi_0^{\varepsilon_{2l-2}} \cdots \phi_0^{\varepsilon_0} \cdot \\ & \mu_0^{M_F(k_0,k)} \tilde{\alpha}_1^{k_{2l}-k_{2l-1}} \cdots \tilde{\alpha}_1^{k_2-k_1} \phi_1^{\varepsilon_{2l-1}} \cdots \phi_1^{\varepsilon_1} \mu_1^{M_D(k_0,k)} \\ & \sum_{s=k_0}^{\hbar_0-1} \left( \tilde{\alpha}^{k_1-\hbar_0} \phi_0^{\hbar_0-s-1} \Upsilon(s) \right) - \sum_{s=\hbar_{2l}}^{k-1} \tilde{\alpha}_0^{k-s-1} \Upsilon(s) \\ & - \sum_{s=k_{2l}}^{\hbar_{2l}-1} \tilde{\alpha}_0^{k-s-1} \phi_0^{\hbar_{2l}-s-1} \Upsilon(s) \end{aligned} \tag{16}$$

Since $\varepsilon_M = \max\{\varepsilon_i\}$ and $1 < \phi_0^{k_{2l}+\varepsilon_{2l}-s-1} < \phi_0^{\varepsilon_M-1}$, under zero initial conditions $\mathcal{V}(\tilde{x}(k_0)) = 0$ and $\mathcal{V}(\tilde{x}(k)) \geq 0$ and according to the Definition 1, it can get

$$\begin{aligned} & \sum_{s=k_0}^{k-1} \tilde{\alpha}_0^{k-s-1} \tilde{\alpha}_0^{\Xi_F(k_0,s)} \tilde{\alpha}_1^{\Xi_D(k_0,s)} e^T(s) e(s) \leq \\ & \chi_0^{\xi_F} \chi_1^{\xi_D} \gamma^2 \sum_{s=k_0}^{k-1} \tilde{\alpha}_0^{k-s-1} \theta_0^{\varepsilon_M-1} \varpi^T(s) \varpi(s). \end{aligned} \tag{17}$$

The accumulated sum of (17) over $[k, \infty)$ is given by

$$\begin{aligned} & \sum_{k=k_0}^{\infty} \sum_{s=k_0}^{k-1} \tilde{\alpha}_0^{k-s-1} \tilde{\alpha}^{s-k_0} e^T(s) e(s) \leq \chi_0^{\xi_F} \chi_1^{\xi_D} \\ & \gamma^2 \sum_{k=k_0}^{\infty} \sum_{s=k_0}^{k-1} \tilde{\alpha}_0^{k-s-1} \theta_0^{\varepsilon_M-1} \varpi^T(s) \varpi(s) \end{aligned} \tag{18}$$

which is equivalent to

$$\begin{aligned} & \sum_{s=k_0}^{k-1} \tilde{\alpha}^{s-k_0} e^T(s) e(s) \leq \chi_0^{\xi_F} \chi_1^{\xi_D} \\ & \theta_0^{\varepsilon_M-1} \gamma^2 \sum_{s=k_0}^{k-1} \varpi^T(s) \varpi(s). \end{aligned} \tag{19}$$

Thus, the closed-loop switched subsystems $\Phi_0$ and $\Phi_1$ are finally shown to be exponentially asymptotically stable and satisfy the exponential $H_\infty$ performance index $\gamma_s =$

$\max \left\{ \sqrt{(\theta_0^{\varepsilon_M} \mu_0)^{\xi_F} (\theta_1^{\varepsilon_M} \mu_1)^{\xi_D} \theta_0^{\varepsilon_M - 1}} \cdot \gamma \right\}$, which completes the proof.

Due to the presence of numerous unknown matrix couplings, it is typically difficult to obtain filter gains from Theorem 1. Then, the linear solvability conditions of the designed filters are proposed in Theorem 2.

**Theorem 2:** Consider the switched subsystems $\Phi_0$ and $\Phi_1$, under DoS attacks with $\tau_F$ and $\tau_D$, scalar $\alpha_i$, $\beta_i$, $\gamma$, $\mu_0$ and $\mu_1$ satisfying $0 < \alpha_i < 1$, $\beta_i > 0$, $\gamma > 0$, $\mu_0 > 1$ and $0 < \mu_1 < 1$. If there exist symmetric positive-definite matrices $\mathcal{P}_{i1}, \mathcal{P}_{i3}$, matrices $\mathcal{P}_{i2}, \mathcal{G}_i, \mathcal{Q}_i, \mathcal{R}_i, \mathcal{A}_{Fi}, \mathcal{B}_{Fi}, \mathcal{C}_{Fi}, \mathcal{D}_{Fi}$, scalar $\gamma, i, j, i \neq j$ satisfying the following conditions

$$\begin{bmatrix} \Pi_i^{11} & \Pi_i^{12} & 0 & \Pi_i^{14} & \mathcal{A}_{Fi} & \Pi_i^{16} & \Pi_i^{17} \\ * & \Pi_i^{22} & 0 & \Pi_i^{24} & \mathcal{A}_{Fi} & \Pi_i^{26} & \Pi_i^{27} \\ * & * & -I & \Pi_i^{34} & \mathcal{C}_{Fi} & 0 & -I \\ * & * & * & -\tilde{\alpha}_i \mathcal{P}_{i1} & -\tilde{\alpha}_i \mathcal{P}_{i2} & 0 & 0 \\ * & * & * & * & -\tilde{\alpha}_i \mathcal{P}_{i3} & 0 & 0 \\ * & * & * & * & * & -\gamma^2 I & 0 \\ * & * & * & * & * & * & -\gamma^2 I \end{bmatrix} < 0, \tag{20}$$

$$\begin{bmatrix} \Pi_{ij}^{11} & \Pi_{ij}^{12} & 0 & \Pi_{ij}^{14} & \mathcal{A}_{Fj} & \Pi_{ij}^{16} & \Pi_{ij}^{17} \\ * & \Pi_i^{22} & 0 & \Pi_{ij}^{24} & \mathcal{A}_{Fj} & \Pi_{ij}^{26} & \Pi_{ij}^{27} \\ * & * & -I & \Pi_{ij}^{34} & \mathcal{C}_{Fj} & 0 & -I \\ * & * & * & -\tilde{\beta}_i \mathcal{P}_{i1} & -\tilde{\beta}_i \mathcal{P}_{i2} & 0 & 0 \\ * & * & * & * & -\tilde{\beta}_i \mathcal{P}_{i3} & 0 & 0 \\ * & * & * & * & * & -\gamma^2 I & 0 \\ * & * & * & * & * & * & -\gamma^2 I \end{bmatrix} < 0, \tag{21}$$

$$\begin{bmatrix} \Omega^{11} & \Omega^{12} & \mathcal{G}_i^T & \mathcal{R}_i \\ & \Omega^{22} & \mathcal{Q}_i^T & \mathcal{R}_i \\ * & -\mu_i \mathcal{P}_{j1} & -\mu_i \mathcal{P}_{j2} \\ * & * & -\mu_i \mathcal{P}_{j3} \end{bmatrix} \leq 0, \tag{22}$$

$$\tau_D < \frac{\varepsilon_M \ln \phi_1 + \ln \mu_1}{\ln \tilde{\alpha}_1}, \tau_F > -\frac{\varepsilon_M \ln \phi_0 + \ln \mu_0}{\ln \tilde{\alpha}_0}, \tag{23}$$

the closed-loop switched subsystems $\Phi_0$ and $\Phi_1$ are exponentially asymptotically stable and and satisfy the exponential $H_\infty$ performance index $\gamma_s = \max \left\{ \sqrt{(\theta_0^{\varepsilon_M} \mu_0)^{\xi_F} (\theta_1^{\varepsilon_M} \mu_1)^{\xi_D} \theta_0^{\varepsilon_M - 1}} \cdot \gamma \right\}$, where $\tilde{\alpha} = 1 - \alpha$, $\tilde{\beta} = 1 + \beta$. $\Pi_i^{11} = \mathcal{P}_{i1} - \mathcal{G}_i - \mathcal{G}_i^T$, $\Pi_i^{12} = \mathcal{P}_{i2} - \mathcal{Q}_i - \mathcal{R}_i$, $\Pi_i^{14} = \mathcal{G}_i^T A_{id} + \mathcal{B}_{Fi} C_d$, $\Pi_i^{16} = \mathcal{G}_i^T B_{1i}, \Pi_i^{17} = \mathcal{G}_i^T B_{2i}$, $\Pi_i^{22} = \mathcal{P}_{i3} - \mathcal{R}_i - \mathcal{R}_i^T$, $\Pi_i^{24} = \mathcal{Q}_i^T A_{id} + \mathcal{B}_{Fi} C_d$, $\Pi_i^{26} = \mathcal{Q}_i^T B_{1i}, \Pi_i^{27} = \mathcal{Q}_i^T B_{2i}$, $\Pi_i^{34} = \mathcal{D}_{Fi} C_d$, $\Pi_{ij}^{11} = \mathcal{P}_{i1} - \mathcal{G}_j - \mathcal{G}_j^T$, $\Pi_{ij}^{12} = \mathcal{P}_{i2} - \mathcal{Q}_j - \mathcal{R}_j$, $\Pi_{ij}^{14} = \mathcal{G}_j^T A_{id} + \mathcal{B}_{Fj} C_d$, $\Pi_{ij}^{16} = \mathcal{G}_j^T B_{1i}, \Pi_{ij}^{17} = \mathcal{G}_j^T B_{2i}$, $\Pi_{ij}^{22} = \mathcal{P}_{i3} - \mathcal{R}_j - \mathcal{R}_j^T, \Pi_{ij}^{24} = \mathcal{Q}_j^T A_{id} + \mathcal{B}_{Fj} C_d$, $\Pi_{ij}^{26} = \mathcal{Q}_j^T B_{1i}, \Pi_{ij}^{27} = \mathcal{Q}_j^T B_{2i}, \Pi_{ij}^{34} = \mathcal{D}_{Fj} C_d$, $\Omega^{11} = \mathcal{P}_{i1} - \mu_i (\mathcal{G}_i + \mathcal{G}_i^T), \Omega^{12} = \mathcal{P}_{i2} - \mu_i \mathcal{Q}_i - \mu_i \mathcal{R}_i^T, \Omega^{22} = \mathcal{P}_{i3} - \mu_i (\mathcal{R}_i + \mathcal{R}_i^T)$.

In addition, if there is a solution to (20)-(23), then the filter gain can be obtained

$$\begin{bmatrix} \mathcal{A}_{fi} & \mathcal{B}_{fi} \\ \mathcal{C}_{fi} & \mathcal{D}_{fi} \end{bmatrix} = \begin{bmatrix} \mathcal{R}_i^{-1} & 0 \\ 0 & I \end{bmatrix} \begin{bmatrix} \mathcal{A}_{Fi} & \mathcal{B}_{Fi} \\ \mathcal{C}_{Fi} & \mathcal{D}_{Fi} \end{bmatrix}. \tag{24}$$

*Proof:* Based on the Project Lemma and the Schur complement Lemma, pre- and post-multiplying (8), one can deduce that (8) and (20) are equivalent. Similarly, pre- and post-multiplying (9) implies that (9) and (21) are equivalent. Theorem 2 is proved.

For the purpose of fault detection, the residual is obtained from the difference between the measured value and its estimated value. Design the following residual estimation function

$$\mathcal{J}_r(k) = \sqrt{\frac{1}{k} \sum_{s=1}^{k} r^T(s) r(s)}. \tag{25}$$

And select threshold value of (25) as

$$\mathcal{J}_{th} = \sup_{\substack{d(k) \in l_2 \\ f(k) = 0}} \mathcal{J}_r(k). \tag{26}$$

Therefore, the fault detection logical relationship is

$$\begin{cases} \| \mathcal{J}_r(k) \| > \mathcal{J}_{th} & Alarm \\ \| \mathcal{J}_r(k) \| \leq \mathcal{J}_{th} & No - alarm. \end{cases} \tag{27}$$

## IV. SIMULATION

This section intends to demonstrate the effectiveness of asynchronous FD strategy for networked UMV under DoS attacks. By choosing matrices $M$, $N$ and $R$ in system (1) as [20]. Let $\alpha_0 = 0.09$, $\beta_0 = 0.05$, $\alpha_1 = 0.11$, $\beta_1 = 0.03$, $\mu_0 = 1.4$, $\mu_1 = 0.45$, $\varepsilon_M = 2$, $\sigma = 1$ and $\gamma = 44$. Then, from (11) the MDADT satisfies $\tau_D < 4.34$ and $\tau_F > 6.60$. The UMV fault detection filter gain under DoS attacks can be calculated by Theorem 2.

To demonstrate the practicability of FD filters designed for networked UMV under DoS attacks, the following simulations are performed to verify it. Firstly, UMV are suffered from thruster faults, external disturbances and DoS attacks. One possible sequences of DoS attacks are depicted in Fig. 1, where 1 denotes that attacks have occurred and 0 denotes the sleep state with no attack. Because of the existence of DoS attacks, which in turn leads to asynchronous switching between the filter and the primary system, then the switching sequence between the filter and the subsystem is shown in Fig. 2.

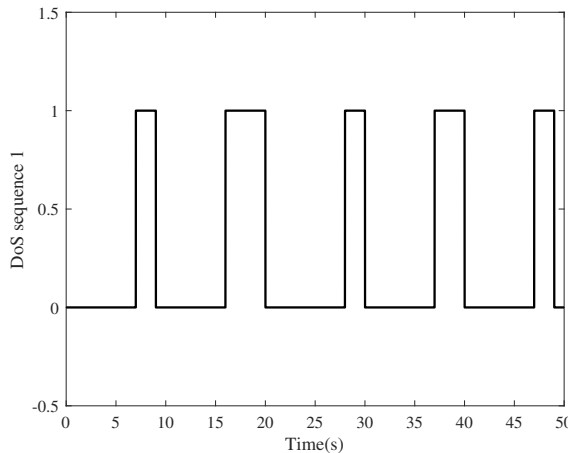

Fig. 1. DoS attacks sequences.

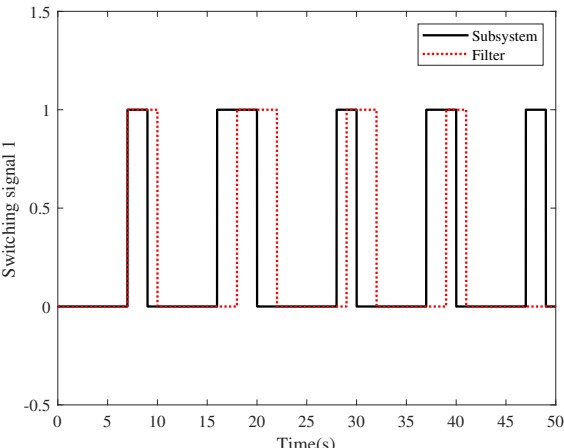

Fig. 2. Switching sequences.

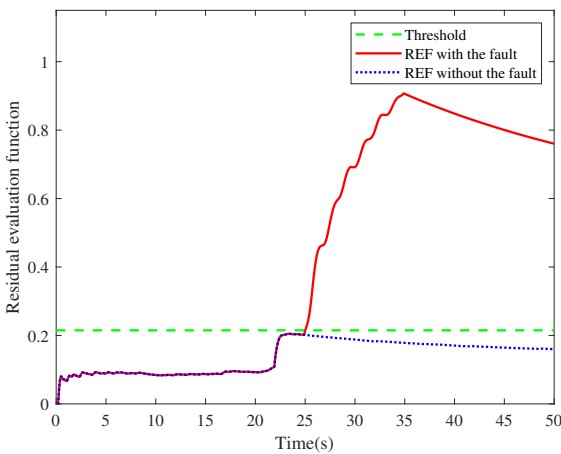

Fig. 4. The REF signal in Case 1.

The external disturbance $d(k)$ is given as the following form

$$d\left(k\right) = \begin{cases} d_1\left(k\right) = 12\sin\left(k\right)\exp\left(-0.15k\right) \\ d_2\left(k\right) = 15\sin\left(0.73k\right), k \in [5,37] \\ d_3\left(k\right) = 9\sin\left(0.2k\right), k \in [11,45] \end{cases}.$$

**Case 1:** Use DoS attacks sequence 1, and the fault signals $f^1\left(k\right)$ takes the following form

$$f^1\left(k\right) = \begin{cases} f_1\left(k\right) = 2\sin\left(0.2k\right) \\ f_2\left(k\right) = \cos\left(0.1k\right) \\ f_3\left(k\right) = 0.8\sin\left(0.15k\right) \end{cases}, k \in [25,35].$$

Under the DoS attack sequence and the faults $f^1\left(k\right)$, the curves of the residual signal $\|r\left(k\right)\|_2$ and the REF signal are depicted in Fig. 3 and Fig. 4, respectively. In the absence of faults, the threshold value is chosen depending on the maximum value of the REF signal: $\mathcal{J}_{th} = 0.215$. When $t = 25.11$s, the fault signal is detected in time.

**Case 2:** In order to further verify the sensitivity of the FD filter to the faults, a fault with a smaller amplitude than case 1 but with the same frequency is selected for verification, and the DoS attack sequence is still used. The fault form of $f^2\left(k\right)$ is shown as follows

$$f^2\left(k\right) = \begin{cases} f_1\left(k\right) = 0.4\sin\left(0.2k\right) \\ f_2\left(k\right) = 0.2\cos\left(0.1k\right) \\ f_3\left(k\right) = 0.16\sin\left(0.15k\right) \end{cases}, k \in [25,35].$$

Under the DoS attack sequence and the faults $f^2\left(k\right)$, the curves of the residual signal $\|r\left(k\right)\|_2$ and the REF signal are depicted in Fig. 5 and Fig. 6, respectively. Fig. 6 indicates that the threshold for fault detection becomes smaller than in *Case 1*: $\mathcal{J}_{th} = 0.067$. And when $t = 25.27s$, the fault signal is detected in time. In contrast to *Case 1*, the residual amplitude and the REF signal are significantly reduced. This shows that the fault amplitude has a non-negligible effect on the system.

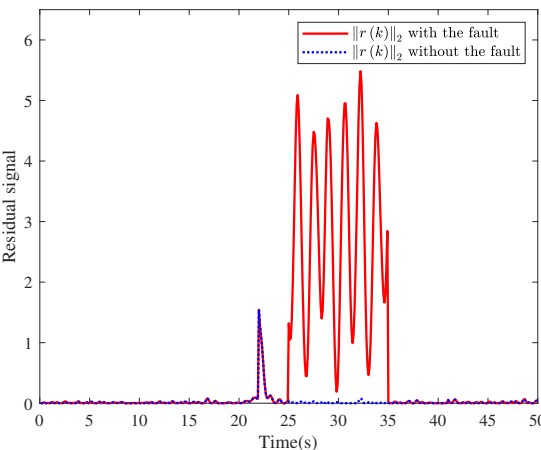

Fig. 3. The residual signal $\|r\left(k\right)\|_2$ in Case 1.

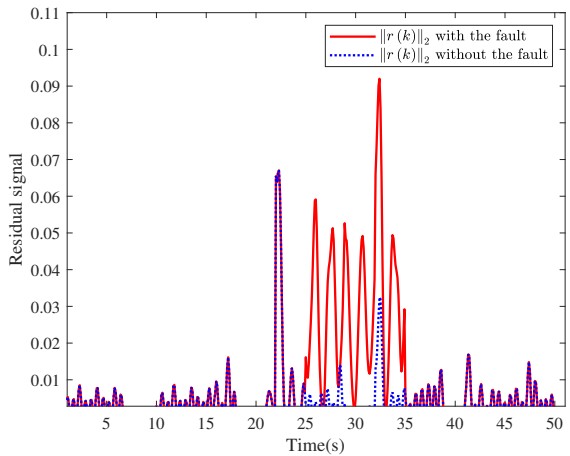

Fig. 5. The residual signal $\|r\left(k\right)\|_2$ in Case 2.

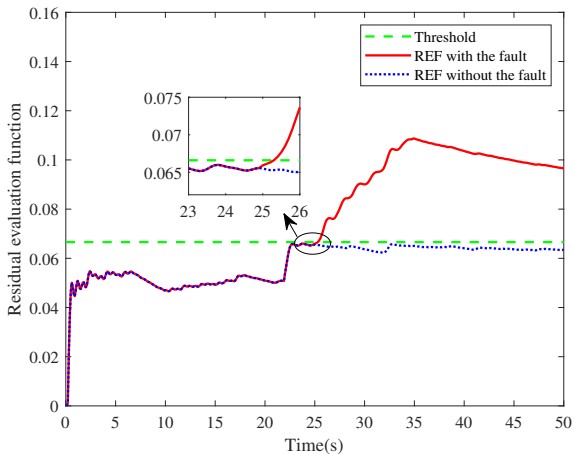

Fig. 6. The REF signal in Case 2.

## V. CONCLUSION

To solve the problem that DoS attacks cannot be detected in time, this paper designs an exponential convergent $H_\infty$ filters based on an asynchronous switched method for UMVs under DoS attacks, which solves the issue that the filters' switching frequently lags behind subsystems in practical applications. On the basis of the MDADT and the PLF, one criterion on the tolerability of the MDADT is derived to maintain exponential $H_\infty$ performance. Sufficient conditions for the designed FD filter to exist are described by LMIs, and the filter gain and the related parameters of MDADT can be derived by solving these LMIs. Finally, the effectiveness of the designed filter is verified by numerical simulation.

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
