# OpenReview forum: "Asynchronous Thruster Fault Detection for Unmanned Marine Vehicles under DoS Attacks"
_IEEE.org/ICIST/2024/Conference — IEEE ICIST 2024 Conference Submission_

### Official Review · Reviewer_4u2C · 2024-08-21
**accept**

**Rating:** 7
**Confidence:** 3

**Review:**

Comment: This paper investigates a thruster fault detection strategy for unmanned marine vehicles (UMVs) subjected to external disturbances and aperiodic Denial of Service (DoS) attacks. The theory is correct and can be accepted after responding the following comments.
(1) More comprehensive literature review is needed to clarify the research gap and research motivation.
(2) Definition 1 and Definition 2 may come from some existing references. The authors should label the corresponding references.
(3) In the end of the conclusions, some research directions are suggested to be added.

---

### Official Review · Reviewer_9YEN · 2024-08-23
**Asynchronous Thruster Fault Detection for Unmanned Marine Vehicles under DoS Attacks**

**Rating:** 7
**Confidence:** 2

**Review:**

This paper investigates a thruster fault detection strategy for unmanned marine vehicles (UMVs) subjected to external disturbances and aperiodic Denial of Service (DoS) attacks. The obtained result is valuable and can be accepted if the following problems can be clarified.
1.	The abbreviated form should be given at the first occurrence in the article, e.g. UMVs, DoS, FD, PLF, etc. In addition, some abbreviations that are not used multiple times in the article are unnecessary.
2.	There is an error in the writing above (15), i.e., (??).
3.	References should be formatted consistently. For example, [4], [18], [20].

---

### Official Review · Reviewer_duFE · 2024-08-24
**This paper investigated a thruster fault detection strategy for unmanned marine vehicles (UMVs) subjected to external disturbances and aperiodic Denial of Service (DoS) attacks. The topic of this paper is interesting. Below is a list of comments that should be taken into account further when revising the paper.**

**Rating:** 7
**Confidence:** 3

**Review:**

1.The contribution of this article should be compared with previous literature, and the basic technical difficulties of this article should be listed? And what methods should be used to solve this problem, emphasizing novelty and technological contribution.
2.In the main results section, the author should provide specific explanations for each figure to ensure the completeness of the article structure.
3.In the conclusion section, solving the issue that the filters’ switching frequently lags behind subsystems in practical applications. Meanwhile, please elaborate on the future plans.

---

### Decision · Program_Chairs · 2024-09-06

Accept (Oral)